# Combined Task and Motion Planning Via Sketch Decompositions

**Primary Keywords:** *(3) Robotics*

## Abstract

The challenge in combined task and motion planning (TAMP) is the effective integration of a search over a combinatorial space, usually carried out by a task planner, and a search over a continuous configuration space, carried out by a motion planner. Using motion planners for testing the feasibility of task plans and filling out the details is not effective because it makes the geometrical constraints play a passive role. This work introduces a new interleaved approach for integrating the two dimensions of TAMP that makes use of *sketches*, a recent simple but powerful language for expressing the decomposition of problems into subproblems. A sketch has width 1 if it decomposes the problem into subproblems that can be solved greedily in linear time. In the paper, a general sketch is introduced for several classes of TAMP problems which has width 1 under suitable assumptions. While sketch decompositions have been developed for classical planning, they offer two important benefits in the context of TAMP. First, when a task plan is found to be unfeasible due to the geometric constraints, the combinatorial search resumes in a specific subproblem. Second, the sampling of object configurations is not done once, globally, at the start of the search, but locally, at the start of each subproblem. Optimizations of this basic setting are also considered and experimental results over existing and new pick-and-place benchmarks are reported.

## Introduction

Combined task and motion planning (Garrett et al. 2021; Lozano-Pérez and Kaelbling 2014), refers to planning problems where a robot manipulates objects in an environment in order to achieve a given goal (Figure 1). The main challenge in TAMP is to integrate planning at two levels effectively: 1) task planning, which requires searching over a combinatorial space to find a sequence of "high-level" symbolic actions; and 2) motion planning, which requires searching for "low-level" paths through the robot's continuous state space. There are two basic approaches for TAMP (Garrett et al. 2021): 1) to search for high-level plans which are subsequently checked for geometric and kinematic feasibility using a low-level solver; and 2) to iteratively generate, typically by sampling, feasible configurations (robot poses, grasps, etc.) which are then used to generate a global sequence of high-level actions.

In this work, we present a novel interleaved TAMP approach to integrate task and motion planning that takes advantage of the powerful language of *sketches* developed recently in classical planning for decomposing problems into subproblems (Bonet and Geffner 2021) which can then be solved by means of effective width-based search algorithms (Lipovetzky 2021). Sketches are collections of rules over state features that can be crafted by hand, expressing domain knowledge (Drexler, Seipp, and Geffner 2021), or can be learned automatically (Drexler, Seipp, and Geffner 2022). The use of sketches for TAMP has two advantages: 1) when a task plan is found to be unfeasible due to geometric constraints, the combinatorial search resumes in a specific subproblem; and 2) the sampling of configurations is performed locally, at the start of each subproblem instead of globally, at preprocessing. While sketches and width-based search have been developed for classical planning, neither one requires declarative action models in PDDL or the like, and only require suitable state features to be defined over the states obtained from a simulator. When the subproblems have a width bounded by $k$, a simple $\text{SIW}_R$ procedure solves the problems in time exponential in $k$ (Bonet and Geffner 2021). For our TAMP tasks, we craft a sketch that yields subproblems of width 1 under suitable assumptions that are discussed.

The paper is organized as follows. We describe first the tasks, review the notions of sketches and width, present the TAMP formulation and the sketches for dealing with three families of pick-and-place tasks, the experimental results, related work, and conclusions.

## Benchmarking TAMP tasks

All the tasks considered as benchmark in this work, depicted in Figure 1, require object manipulations through pick-and-place actions in cluttered scenarios and have the same high-level goal of ensuring that no object remains misplaced, which is a goal that fits to many robotic manipulation tasks. The first two tasks are taken from a TAMP benchmark (Lagriffoul et al. 2018) that aims to be a standard on this field. The third task is inspired in the classical *Blocks World*.

*Sorting Objects*— A robot must arrange different blocks standing on different tables, based on their color. The goal constraints are that all $N$ blue blocks must be on the left table and all $N$ green blocks must be on the right table. There are also $2N$ red blocks, acting as obstacles for reaching blue and green blocks, whose goal position is free. The robot is allowed to freely navigate around the tables, while keeping

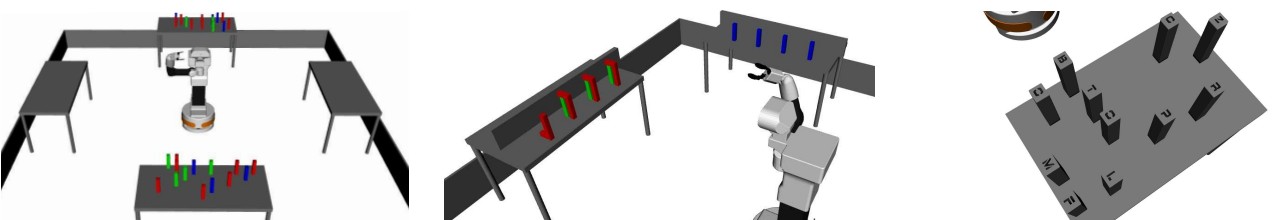

Figure 1: Examples of TAMP considered in this work: *Sorting Objects* (left), *Non-Monotonic* (middle), and *Words* (right).

within the arena, picking and placing the blocks at the tables. The proximity between the blocks forces the planner to carefully order the operations, as well as to move red blocks out of the way without creating new obstructions, i.e., blocking objects. Thus, this problem requires to move many objects, sometimes multiple times.

*Non-Monotonic*— A robot must move three green blocks standing on a table to their corresponding goal position on another table. At the initial state, there are four red blocks obstructing the direct grasp of the green blocks and there are also four blue blocks obstructing the direct placement of the green blocks at their target position. In any case, the red and blue blocks must end up being at the same initial locations. The robot is allowed to freely navigate around the tables, while keeping within the arena, picking and placing the blocks at the tables. The goal condition on blue and red blocks requires to temporarily move them away and bring them back later on (non-monotonicity), to solve the task.

*Words*— A robot must arrange anywhere on a table some blocks, each one labelled with a letter (possibly repeated), to build a target word, e.g., TAMP and ROBOT. Then, the goal positions of each block are not absolute but relative to the positions of other blocks, and blocks can share goal positions if a letter is repeated in the goal sequence. Besides, the available free space on the table is limited, and thus there are obstructions between blocks. This modification is intended to make the coupling of the geometric and symbolic reasoning more challenging. Thus, a block is well-placed if, in a given state, it belongs to the largest valid sequence (by *valid* we mean that the sequence is within the goal sequence and located far enough away from the edges of the table to allow the whole sequence to be completed).

## Combined Task and Motion Planning

We assume the reader is familiarized with the framework of width-based planning and sketches. A background section, providing a review of these concepts, is available in the supplementary material.

We consider pick-and-place problems where, at a high-level, a mobile robot with an arm can perform three actions: picking up an object from a surface, leaving it on a surface, and moving to another place (dragging any grasped object). Underlying this action-space there is a complex, lower-level problem involving a continuous state-space (position values of robot joints, robot and object locations and spatial constraints). The formulation exploits a particular width-based task-planner based on the $\text{SIW}_R$ algorithm (Bonet and Geffner 2021) guided by a simple and general sketch $R$ with

a handful of chosen state-features that decompose the problems into subproblems solved by a linear-time search guided by IW(1) (Lipovetzky and Geffner 2012). This task-planner calls a motion planner for checking the feasibility of high-level actions and for driving the joints. Two characteristics of the proposed integration is that the sampling required for mapping the high-level action schemas into ground actions is done at the level of subproblems, and that "backtracking", namely failure of high-level subplans, is carried out also at the subproblem level. There is no "deep backtracking", the $\text{SIW}_R$ search proceeds forward from the goal, one subgoal at a time by means of the linear-time IW(1) procedure, and it is the width $w$ of the sketch over the family $\mathcal{Q}$ of pick-and-place tasks that ensures that this search will be complete; namely, when $w = 1$.

## Problem Formulation

The class of TAMP tasks that we address can be characterized as a tuple $T = (S, s_0, G, Act, A, X, N)$ where:

- $S$ represents the continuous states with all relevant information, $s_0$ the initial state and $G$ the set of goal states;

- $Act$ is the set of high-level action schemas;

- $A$ is the *sampling function* that given a state $s$ and the set of action schemas $Act$ produces the set of *sampled grounded actions* $A(s)$ that are relevant in $s$ (although not necessarily executable in $s$). Thus, $A(s)$ limits the set of states $S$ that the planner can potentially visit given $s$;

- $X$ is an *executability function* that maps ground actions $a \in A(s)$ and states $s \in S$ into *motion plans* $X(a, s) = \rho$ that implement the action $a$ in $s$ at the low level, if the ground action is feasible; if not, $X(a, s) = \bot$;

- $N$ is the state-transition function that yields the next state $s'$ when the motion plan $X(a, s) = \rho \neq \bot$, is executed in $s$; i.e. $s' = N(s, \rho)$ when $\rho \neq \bot$.

A *plan* or *solution* for a tuple $T$ is a sequence of ground actions $a_0, \ldots, a_n$ such that: 1) $a_i \in A(s_i)$, $i = 0, \ldots, n$; 2) $s_{i+1} = N(s_i, \rho_i)$ for $\rho_i = X(a_i, s_i)$, $\rho_i \neq \bot$, $i = 0, \ldots, n$; and 3) $s_{n+1} \in G$. Starting with $s_0$, the state progresses according to low-level motion plans and the given dynamics, and this progress must end in a goal state.

In the search, the main bottleneck is in the execution function $X(a, s)$ that is expensive as it involves run-time calls to a motion planner. Note also that one special case of the above formulation arises when the configuration sampling that leads to the finite set $A(s)$ of ground actions is done only once at pre-processing. In such a case, $A(s) = A(s_0)$

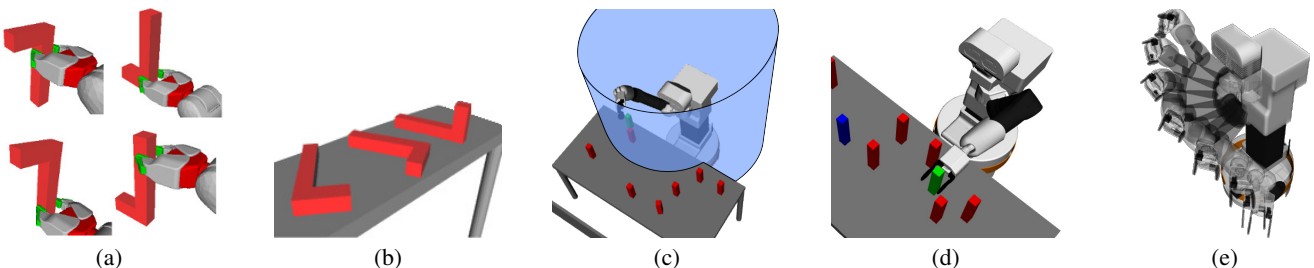

Figure 2: Different grasps (a), SOPs (b), and calls to *InArmWorkspace* (c), *InverseKinematics* (d), and *MotionPlan* (e) functions.

for all states. Other variations do one sampling per subproblem; in such a cases $s$ "borrows" the set of ground actions from other states $s'$ considered before as $A(s) = A(s')$.

### State-Space $S$, start state $s_0$ and goal states $G$

On the one hand, the state space $S$ is continuous and comprises the robot configuration-space (i.e. the position value of the robot joints), the robot pose (i.e. position and orientation in Cartesian space) together with the pose of the movable objects. Formally, $S \in \mathbb{R}^d \times \text{SE}(3) \times \cdots \times \text{SE}(3)$, where $d$ is the number of degrees of freedom of the robot and $\text{SE}(3)$ is the group of homogeneous transformations, one for the robot and as many as objects. Note that $S$ is what the motion-planner "sees" and that low-level discrete information can be inferred from it (e.g. knowing that an object is grasped by checking that its position coordinates are within the gripper fingers). On the other hand, the initial state $s_0$ involves the starting configuration of the robot and the objects and the set $G$ is the set of states $s$ that imply that all misplaced objects are in their goal positions/regions.

### Action Schemas $Act$

Three high-level action schemas in $Act$ are considered in $T$:

- $\text{Pick}(b, o, g)$ actions that, when valid, allow the robot to pick up, from a given robot location $b$, a specific object $o$ using a specific grasp $g$ (see Figure 2a), as long as the robot is not already holding any object.

- $\text{Place}(b, o, p, \sigma)$ actions that, when valid, allow the robot to place, from a given robot location $b$, a particular object $o$ (if it is the one being held), in a given placement $p$ using a given SOP $\sigma$ (Stable Object Pose, see Figure 2b).

- $\text{Move-Base}(b, b')$ actions that, when valid, allow the robot to navigate from its current position $b$ to a different one $b'$, dragging the held object in case there is one.

Note that actions are grounded when all the related parameters are fully specified. Thus, the action space comprises all the possible actions that could be obtained by the combination of the possible values of these parameters.

### Ground Actions $A(s)$ through Sampling

Given a state $s$, the sampling function $A$ produces a set $A(s)$ of sampled grounded actions by sampling values for all the

*Act* parameters. Note that the size of $A(s)$ increases exponentially with the number of movable objects and polynomially with the sampled parameters. The parameter values are randomly sampled seeking both a uniform density and maximum coverage of the parameter-space, taking into account that: 1) the set of sampled placements $p$ must contain the poses of the movable objects at $s$ and their goal position(s) in $G(s)$; 2) the placements $p$ can only be on the tables; and, 3) the robot locations $b$ must be at distance from a table less than the radius of the arm's reach (outside this region the robot cannot manipulate any object and, besides, this useless base location increases the problem complexity). This promotes richness of representation and the sampled $A(s)$ potentially generate (if executed) states where objects are spaced and, thus, geometric constraints are easier to satisfy.

### Executability and Transition Functions $X$, $N$

In a given state $s$, the executability function $X$ maps a grounded action $a \in A(s)$ to a motion plan $\rho = X(a, s)$, i.e. a trajectory in the robot joint-space that respects its capabilities and does not involve self-collisions or collisions with the environment. The action is not be feasible if $\rho = \perp$. For these operations and checks, the motion planner MoveIt! is used (Coleman et al. 2014) to compute the following three functions sequentially:

1. *InArmWorkspace*: Returns `true` when a given target (e.g. an object to interact with) is within the robot arm's reach, and `false` otherwise (see Figure 2c).

2. *InverseKinematics*: Returns `true` only when a valid non-collision robot joint configuration is found, to place the robot gripper at a target (see Figure 2d), using a general robot-agnostic iterative Inverse Kinematics (IK) solver algorithm (Chiaverini, Siciliano, and Egeland 1994). Notice that if the target is not within the arm workspace, i.e. *InArmWorkspace* fails, an IK solution does not exist, and *InverseKinematics* will fail too.

3. *MotionPlan*: Returns `true` only when a valid motion plan $\rho \neq \perp$ is found, to place the robot gripper in a target and perform the corresponding action (see Figure 2e), using a motion-planning algorithm. Here, RRT-Connect is used (Kuffner and LaValle 2000), which is probabilistically complete (if there exists a motion plan, it will find it provided sufficient time). Note that if there is no valid robot configuration to grasp an object, i.e. *InverseKinematics* fails, a motion-plan solution does not exist and, hence, *MotionPlan* will fail after a timeout.

The state-transition function $N$ returns the last state $s'$ of the motion plan $\rho \neq \perp$.

## Solving TAMP: Sketch Decompositions

We will search for plans that solve a given TAMP task $T = (S, s_0, G, Act, A, X, N)$ by means of the the $\text{SIW}_R$ algorithm that uses a domain-dependent sketch $R$ to decompose the problem into subproblems, and the IW search algorithm for solving the subproblems. If the width of the sketch $R$ for the target class of problems $\mathcal{Q}$ is bounded, then $\text{SIW}_R$ will solve the problems in polynomial time, and moreover, if the sketch width is 1, $\text{SIW}_R$ will solve the subproblems, greedily with the IW(1) algorithm, in linear time. For making this possible, the features used in the sketch $R$, and the features used in the IW algorithm, need to be chosen carefully, assuming that the class of problems $\mathcal{Q}$ can be solved in polynomial time. For the TAMP setting, we provide the sketch and the features by hand, although they could be learned from small instances (Drexler, Seipp, and Geffner 2022), provided suitable *state languages* (in planning, the state languages is given by the domain predicates).

The set of features $F$ is given by two sets $F = (F_H, F_D)$ of Boolean and numerical state features (functions): the first set $F_H$ is used only in the IW searches invoked by the $\text{SIW}_R$ algorithm and they are just Boolean features; the second set $F_D$ is used in the sketch $R$ for expressing the decomposition of a CMTP problem $T$ into subproblems, and uses both Boolean and numerical features. The numerical features *counters* take non-negative integer values. The abstract state space that is searched is actually given by the possible values of the features. It is assumed that the features for the class of problems $T$ are rich enough to distinguish goal states from non-goal states.

### Features $F_H$ for Searching Subproblems

Given a sketch with rules $C \mapsto E$, the subproblem to be solved at a state $s$ is given by a problem that is like $T$ but with initial state $s$ and goal states $G_R(s)$ given by the original goal states of the problem $T$, along with the "subgoal" states $s'$ such that the state pair $[s, s']$ satisfies a sketch rule $C \mapsto E$ in $R$; namely, $s$ makes the condition $C$ true, and the transition from $s$ to $s'$ makes the effect expression $E$ true (for instance, a feature is decremented). These subproblems, denoted as $T[s, G_R(s)]$, are solved by an IW(1) search where the "atoms" in a state $s$ are given by the Boolean search features $F_H$. The IW(1) search over the subproblem $T[s, G_R(s)]$ thus proceeds as a breadth-first search from $s$ where newly generated states that do not make a feature in $F_H$ true for the first time. The search succeeds when a state $s'$ in $G_R(s)$ is found, else it fails. The applicability of an action $a \in A(s)$ is checked by evaluating the executability function $X(a, s)$: if it results in a motion plan $\rho$, the state resulting from the action $a$ in $s$ is set to $s' = N(s, \rho)$.

The set $F_H$ of search features considered capture the robot and objects poses and they are called $\texttt{robot-at-}p$ and $\texttt{object-}o\texttt{-at-}p$, for poses $p$ and objects $o$. It can be shown that if the subproblems $T[s, G_R(s)]$ generated by the sketch $R$ can all be solved by moving an object from some original pose to another pose, the width of the subproblems with these features $F_H$ is 1, and thus they can be solved efficiently by IW(1), while generating a number of nodes that is linear in the number of objects, sampled configurations, and number of features in $F_H$.

### Sketch, $\text{SIW}_R$ and Problem Decomposition

Starting with a state $s$, the $\text{SIW}_R$ algorithm carries an IW search from $s$ until a state subgoal state $s'$ in $G_R(s)$ is found. If $s'$ is a goal state of the problem, the $\text{SIW}_R$ exits successfully, else it repeats the process from $s := s'$. The role of the sketch $R$ is to define the set of subgoal states $s' \in G_R(s)$ as the goal states of the problem, and the subgoal states $s'$ that along with $s$ satisfy a sketch rule.

For the different types of pick-and-place tasks to be addressed, a hand-made sketch $R$ with four rules involving five features is used. The five features are:

- $H$: a Boolean feature that is $\texttt{true}$ if the robot is holding an object, and $\texttt{false}$ otherwise.
- $I$: a Boolean feature that is $\texttt{true}$ if there exists a goal pose of the grasped object not blocking the pick/place of any other misplaced object, and $\texttt{false}$ otherwise.
- $m$: the number of *misplaced* objects, defined as the objects that are standing (i.e. not held) outside their goal position/region (if the object has no associated goal, it is never misplaced) or are held and their goal position/region is blocked. The goal positions/regions depend on the problem and they can be defined in absolute coordinates (e.g. *Sorting*) or relative coordinates (e.g. *Words*).
- $u$ and $v$: respectively $\min(\alpha_i + \beta_i)$ and $\sum \alpha_i$, with $\alpha_i$ being the minimum number of objects blocking the $i$-th misplaced object (i.e. preventing it from being picked up from its current position and put down in its goal position/region) and $\beta_i$ the minimum number of misplaced objects blocked by the goal position(s) of the $i$-th misplaced object. During the search for such a minimum value, it is considered the best robot location and grasp for the *pick* action and the best robot location, object placement and SOP for the *place* action in the goal zone, while maintaining consistency (i.e. using the same grasp for the considered *pick* and *place* pair).

Using these features, the goal in all the tasks is expressed as $\neg H$ and $m = 0$ (no misplaced objects and no held object).

The sketch $R$ used is made up of four sketch rules explained below. Recall that $m?$ indicates that it does not matter how $m$ changes, and a non-mentioned feature must preserve its value.

- $\{\neg H, m > 0, u = 0\} \mapsto \{H, F, m\downarrow, u?, \neg v\uparrow\}$: if there is some misplaced object that can be picked up and placed down directly, pick it up. The expression $\neg v\uparrow$ means that $v$ can change but not increase.
- $\{\neg H, m > 0, u > 0\} \mapsto \{H, I?, m?, u?, v\downarrow\}$: if there are misplaced objects but they cannot be picked up directly, pick up an obstructing object. Notice that, even when $u$ were inaccurately estimated, picking an unexpectedly-accessible misplaced object would also satisfy this rule (which is not an inconvenience at all).

- $\{H, \neg I\} \mapsto \{\neg H, I?, m?\}$: if robot is holding an object that cannot be placed in goal without blocking misplaced objects, place it somewhere else without creating new interferences with the objects that are still misplaced.
- $\{H, I\} \mapsto \{\neg H, I?\}$: if the robot is holding an object, which would not block other misplaced object once placed if it had a goal associated with it, enforcing to put it down but not in a wrong position and without disturbing the access to the pending misplaced objects.

The sketch structures the problem into subproblems but it is not a policy; the subproblems need to be solved by search, but this is a search that is done efficiently by IW(1) in $\text{SIW}_R$. For example, the rules do not say how the base of the robot should move, or which grasp to use. Part of these details are filled in by the search (e.g. robot base moves) and the others by the motion planner (e.g. grasps). Besides, note that while the approach is general, the given sketch is suited for (a large class of) pick-and-place tasks. In particular, TAMP problems involving other "high-level actions" would require different sketches, e.g. a robot-cook will need actions for heating ingredients.

**Formal Properties.** The effectiveness of the $\text{SIW}_R$ procedure on the TAMP tasks $T$ considered below depends on the problem decomposition that follows the choice of the sketch $R$ (and its features $F_D$), and the power of the subproblem search by means of IW(1) with the given search features $F_H$. Two properties that would guarantee that $\text{SIW}_R$ using IW(1) solves the problem $T$ are: 1) sketch termination (Bonet and Geffner 2021), and 2) sketch width of 1 for this family of tasks. We prove the first and provide a rationale for the second, that in the TAMP setting involves conditions that cannot be captured with precision.

**Sketch termination.** It must ensure that moves from state $s$ to subgoal state $s' \in G_R(s)$ cannot go on for ever. For this, we show that the four sketch rules can be used to generate subgoals a finite number of times only. First, note that feature $v$, that captures roughly total number of "interferences" towards the goal, is decreased in the second sketch rule $r_2$ but is not increased in any of the four rules (no expression effects $v\uparrow$ or $v?$). This means that rule $r_2$ can be used for subgoaling a finite number of times only as counters cannot become negative. Once that rule $r_2$ is excluded from those that can be "followed" an infinite number of times, a similar argument can be made about rule $r_1$ where the counter $m$ is decreased. In this case, rule $r_3$ may increase $m$ as it contains the effect expression $m?$, yet this rule cannot be followed after $r_1$ because $r_1$ makes $I$ true and $r_3$ requires $I$ false. So infinite "executions" of rule $r_1$ can only involve infinite executions of rule $r_4$, as rule $r_2$ cannot be executed infinitely often, and since $r_4$ cannot increase the value of $m$, $r_1$ cannot be executed infinitely often, but then neither $r_3$ nor $r_4$ that make $H$ false but require $H$ true in the antecedent.

**Sketch width 1.** The width of the presented sketch for the class of pick-and-place problems $T$ with search features $F_H$ can be shown to have width 1 under the following conditions: 1) the counters $m$, $u$, and $v$ represent all interferences affecting misplaced objects; 2) there is reachable space for placing objects without causing new interferences (increases in $u$ and $v$); 3) every object can be reached, possibly after moving other objects out of the way, 4) the ground parameters (sampling) of the relevant ground actions in the subproblem are sampled. These are strong assumptions but they are needed for explaining the power and scope of the proposed method. It is important that the sampling is done in the context of a simple subproblem, and does not need to be done once for the whole problem. We show then that in every state $s$, the problem of reaching a subgoal state $s'$ $G_R(s)$ has width 1. The four sketch rules $r_i : C_i \mapsto E_i$ have disjoint antecedents $C_i$, so we just need to consider four cases, where $s$ makes $C_i$ `true`, and $s' \in G_{r_i}(s)$ where the set of rules $R$ is replaced by the single rule $r_i$, $i = 1, \ldots, 4$.

*Rule 1.* If $s$ satisfies $C_1$, the subproblem $T[s, G_{r_1}(s)]$ involves picking up a misplaced object with no obstructions, whose goal is free and can be occupied without causing obstructions. The rule ensures that there is one such object and that picking up an object that does not comply with these conditions will not lead a state $s'$ in $G_{r_1}(s)$. Therefore, there is a plan to solve this subproblem in which the robot moves certain times and the object is picked up. If this plan is optimal, the sequence of robot base positions, until the pick up, captured by `robot-at-`$p$ features, followed by the object configuration after the pick up, captured by `object-`$o$`-at-`$p$ features, yields chain of "atoms" $t_0, \ldots, t_k$ that is *admissible*; namely, $t_0$ is `true` in $s$, optimal plans for $t_i$ can all be extended into optimal plans for $t_{i+1}$, and optimal plans for $t_k$ are optimal solutions to the subproblem $T[s, G_{r_1}(s)]$. Hence, IW(1) running with the proposed $F_H$ will solve this subproblem optimally (Lipovetzky and Geffner 2012; Bonet and Geffner 2021).

*Rule 2.* If $s$ satisfies $C_2$, the subproblem $T[s, G_{r_2}(s)]$ involves picking up an object that is obstructing misplaced objects, for moving it "out of the way". The subproblem can be solved in the same way by `Move-Base` actions, followed by a `Pick`, giving rise to a similar admissible chain of atoms that establishes that the subproblem has width 1 and is solved optimally by IW(1).

*Rule 3.* If $s$ satisfies $C_3$, the subproblem $T[s, G_{r_3}(s)]$ involves placing the object held "out of the way", without increasing the number of interferences. The optimal plan for the subproblem may involve a number of robot moves followed by placing down the held object. In this case, the "atoms" `object-`$o$`-at-`$p$ that are made `true` in an optimal plan for the subproblem constitute an admissible chain that proves that the subproblem has width 1.

*Rule 4.* If $s$ satisfies $C_4$, the subproblem $T[s, G_{r_4}(s)]$ involves placing the object held, that appears in the goal, whose target is not obstructed and that placed in that target will not cause obstructions to misplaced objects. Once again, the conditions ensure that the subproblem has a solution, and potentially involves `Move-Base` actions followed by a `Place` action as well. The "atoms" `object-`$o$`-at-`$p$ that are made `true` in an optimal plan for the subproblem constitute an admissible chain, that proves that the subproblem has width 1 (under the above, general conditions).

## Optimizations

Here, we introduce implementation details, including used approximations and optimizations. There are two sources of approximation: 1) Representation of continuous variables through sampling and, 2) Relaxing the Sketch-Features computation. Furthermore, two search optimization mechanisms are introduced that focus on minimizing the efforts on geometric validation of the actions.

**Adaptive sampling and Probabilistic completeness.** Since the sketches allows decomposing the problem into subproblems, we use a different state-space adapted to each subproblem. This enables working with reduced tractable state-spaces that represent well each subproblem scenario. Note that any of these specific discretizations is not sufficiently-rich to represent well the whole problem world and yet, combined, they allow finding a solution to the overall problem. Besides, the sampling density of the $Act$ parameters is increased (and the search is restarted) when an attempt to reach a subgoal fails, to obtain new values that can help to find a subproblem solution. Thus, the full approach is probabilistically complete in the sense that (a) all object locations and configurations are susceptible to be sampled; (b) when a subsearch fails, the same subsearch is reattempted but with more sampled object locations.

**Sketch-Features computation.** The computation of the features $I$, $u$ and $v$ would imply complex inverse-kinematics and collision-checking computations. To speed up the process, these type of computations are avoided and approximations are used instead in the approach implementation. In particular, instead of finding the robot configuration that causes the fewest collisions for a given pick/place of a misplaced object, we overestimate this number by looking at how many objects have their center of mass within a 3D region containing all possible inverse kinematics solutions. This region is easy to compute and is implicitly defined considering the object and robot poses and the grasp to be used.

**Lazy action-validation.** To speed-up the search, the action validation is relaxed until a subplan is found. In particular, the third step of the executability function $X$ (i.e. *Motion-Plan*) is only checked for the actions in a potential subplan. Hence, graph edges are "provisional" until complete checking and we must keep all the discovered edges. Thus, a node may have more than one parent node (although the one implying a lowest cost to reach the start is the one acting as the parent). With this lazy approach, the most time consuming action-validation step is performed only on those actions that are part of a potential solution plan. If all the plan actions are successfully validated, a completely valid subplan is returned. Else, the unfeasible action is discarded and the corresponding graph edge is removed. However, the child node is kept as long as there is an edge supporting it.

**Incremental IW($k$).** Instead of restarting completely the IW($k$) subsearch when a potential subplan is invalidated by the motion planner, we save the search done until then and resume the search. For this, the novelty management is adapted such: 1) All the features associated to each generated node are recorded (it could still be pruned, if it does not introduce the required novelty); 2) A feature is supported by one main node but it may have newer nodes as backup candidates; 3) When a node has to be discarded because it has been disconnected from the root node during a final action-validation, this node is removed as support of its related features; 4) If a feature loses all of its supporters it is removed from the novelty table; and, 5) A pruned node (because of not passing the novelty check) could be recovered if it is the next backup candidate after removing a support node.

The resulting algorithm after applying the explained optimization is named in this work Lazy Serialized Incremental Iterated Width with Sketches (**Lazy-SIIW**$_R$).

## Related Work

Our approach to TAMP interleaves task and motion plans like (Garrett et al. 2021). Two other related approaches are *sequence-before-satisfy*, that first obtain fully-symbolic potential plans and then solve the geometric constraints (Garrett, Lozano-Pérez, and Kaelbling 2020), and *satisfy-before-sequence* (Akbari, Muhayyuddin, and Rosell 2016), that first solve the geometric problem and then find action sequences that use those values.

Our work is also related to Planet (Thomason and Knepper 2022), which interleaves task and motion planning through a flexible sampling-based approach. Similarly, Planet avoids sampling in the full Cartesian space. However, Planet uses a composite space including symbolic and continuous components, and defines an explicit embedding of the symbolic state into the continuous space. Furthermore, Planet uses a heuristic to guide the sampling. Instead, we consider a blind but focused search, IW(1), to solve each subproblem, and the subproblem to solve next follows from the sketch. An interleaved approach using sampling mechanism of predefined primitives to obtain (only valid) continuous values in the node expansion is proposed in (Ajanović et al. 2023), where a feasibility map is proposed to enable approximated models for motion primitives generation.

Most related to our approach is the work of Ferrer-Mestres, Francès, and Geffner (2017) which uses Best-First Width-Search (BFWS) on a pre-discretized state-space and restricts the sampling to a template-grid (kept for the whole search). BFWS uses an extension of PDDL that accommodates procedures and state constraints (Geffner 2000). State constraints represent implicit action preconditions to discard spatial overlaps. Procedures are used for testing and updating robot and object configurations. Other works extend PDDL in different ways. For example, PDDLStream (Garrett, Lozano-Pérez, and Kaelbling 2020) incorporates sampling procedures in PDDL that allow a planner to reason about conditions on the inputs and outputs of a conditional generator while treating its implementation as a black box.

Using sketches to address a collection of tasks have parallelisms with generalized planning (Srivastava, Immerman, and Zilberstein 2008). Like general policies (Bonet and Geffner 2018), sketches are general and not tailored to specific instances of a domain, but unlike policies, the feature changes expressed by sketch rules represent sub-goals that do not need to be achieved in a single step. A methodology for learning sketches in classical planning tasks has been

introduced in (Drexler, Seipp, and Geffner 2022). Learning features, abstractions, and generalized plans from a few examples in continuous TAMP problems has been proposed by Curtis et al. (2022).

The idea of lazy evaluation is known to reduce planning time for search-based planners and has been applied recently in several works, e.g, by postponing applicability checks for successor states performing geometric queries (Dornhege, Hertle, and Nebel 2013), by deferring motion sampling until an action skeleton is found (Khodeir, Sonwane, and Shkurti 2022), or by solving shortest path problems as needed (Dellin and Srinivasa 2016).

An alternative formulation leverages nonlinear optimization to jointly compute a motion that satisfies geometric and physical constraints (Ortiz-Haro et al. 2022). Contrary to our approach, these methods operate on the full planning sequence and do not explicitly exploit the subproblem decomposition. Direct comparison with our approach is difficult, as they require the formulation of the optimization problem, e.g. the logic-geometric program, and a full PDDL problem-description.

Hierarchical planning approaches also express and exploit problem decompositions (Wolfe, Marthi, and Russell 2010; Kaelbling and Lozano-Pérez 2011). In this cases, the decomposition is expressed in hierarchical manner and not by means of sketches. The key difference between these hierarchical approaches and the sketch-based approach is the role and scope of the search. In the proposed sketch-based approach, the subproblem search is done in linear time, by means of the IW(1) algorithm, and this guaranteed to be complete if the sketch width is 1. The role and scope of the search in hierarchical approaches in both robotics and planning is less clear: either the domain knowledge must express a full strategy for solving the problems which does not require any search, or else, the search can easily get lost. More recent works make use of hierarchical models and combine them with learning (Patra et al. 2020).

Other related methods make flexible use of external predicates and functions for feasibility checks (Erdem, Patoglu, and Saribatur 2015; Erdem et al. 2011). These methods strongly rely on the problem description, which is modified during the search integrating additional domain specific information from the motion-level. Learning-driven approaches can be used to guide the solution of TAMP problems. (Kim and Shimanuki 2020) proposed learning action-value functions for speeding up the discrete part of the search in TAMP problem. Another learning approach, introduced in (McDonald and Hadfield-Menell 2022), consists on training a policy to imitate a TAMP solver's output. The obtained feed-forward policy is used to solve tasks from sensory data and to supervise the training an asynchronous distributed TAMP solver for imitation learning is used.

## Experimental Results

We evaluate our method in the three tasks[1] presented initially with the following objectives in mind: to provide empirical evidence of the theoretical results, to analyze how the

[1]Supplementary material includes accompanying videos.

method scales with the problem complexity, to quantify the impact of the lazy action-validation, and to compare it with related approaches. We compare the following methods:

- **Lazy-SIIW**$_R$: our approach with lazy action-evaluation, called Lazy Serialized Incremental IW with Sketches.
- **SIW**$_R$: our approach without lazy action-evaluation.
- **BFWS** (Ferrer-Mestres, Francès, and Geffner 2017).
- **Planet** (Thomason and Knepper 2022).
- **PDDLStream** (Garrett, Lozano-Pérez, and Kaelbling 2020). We extended the existing implementation of PDDLStream planners. Details of these extensions are provided in the supplementary material.

Table 1 shows the results. Both Lazy-SIIW$_R$ and SIW$_R$ are able to solve all problem instances. This confirms that the assumptions regarding sketch termination and sketch width are satisfied for these benchmarks, and shows that the method remains complete even in the presence of approximations. Unlike all other methods, Lazy-SIIW$_R$ requires less time for *computing* the plans than for *executing* them, which shows the benefits of the lazy action-evaluation.

Lazy-SIIW$_R$ and SIW$_R$ also scale up well, despite the exponential increase in the state-space due to an increase in the number of objects. Compared to BFWS, Lazy-SIIW$_R$ is always more efficient both in planning and execution time. Without the lazy evaluation of constraints, SIW$_R$ only performs comparably to BFWS. PDDLStream planners and Planet are only comparable to Lazy-SIIW$_R$ for simple problems, but they exceed the time limit as soon as the complexity of the problems increases.

To understand better how the lazy action-evaluation affects the performance, we analyze how the computation is distributed between the different steps involved in the geometric validation of each action. Table 2 shows illustrative profiling values for the three steps. Note that they act in increasing order of computational load and decreasing order of relative discriminative power (i.e., rejection ratio over the non-filtered actions in the previous step). This permits to efficiently discard most of unfeasible actions. Decomposing the action-validation enables delaying the computation of the most expensive step (*MotionPlan*). Thus, the complete validation is only computed on those actions that are part of a potential plan. Note that satisfying the first two steps almost ensures that the action is valid.

Another important metric is the quality of the produced plans. Although optimality can not be guaranteed, the proposed algorithm performs well in avoiding unhelpful actions (i.e. obtaining near-optimal solutions). This follows from the rules in the sketch that discourage moving objects that are not misplaced and do not block misplaced objects. Hence, after moving these objects, often other goal objects became accessible without needing to move non-goal objects. Indeed, when comparing the results with the other approaches, plans containing fewer actions are obtained for the same problems with the proposed approach.

High-cluttered problems are the most challenging. On these problems, the goal objects can be obstructed by many objects, which must be moved away, and there is a limited

Table 1: Average results over ten runs in the benchmarking problems in an Intel® Core™ i7-10610U CPU at 1.80 GHz, with 16 Gb of RAM, on Ubuntu 20.04.4 and ROS Noetic. †Within 30 min maximum planning time or maximum memory exceeded. ⊥Considering that the path execution does not start until the path has been completely planned. *Including also the pre-processing time, for a fair comparison.

| Prob. | #Tables | #Objects | #Goal objects | Clutter level | Planner | Success ratio† | Planning time | Execution time | Total time⊥ | #Expanded nodes | #Sub plans |
|---|---|---|---|---|---|---|---|---|---|---|---|
| Sorting Objects | 1 | 20 | 2 | High | **Lazy-SIIW$_R$** | 100% | 3.16 min | 3.67 min | 6.83 min | 60 | 14 |
| | | | | | SIW$_R$ | 100% | 13.05 min | 3.73 min | 16.78 min | 54 | 14 |
| | | | | | BFWS | 100% | 5.25 min* | 5.91 min | 11.16 min | 63.3k | 1 |
| | 3 | 2 | 2 | Low | **Lazy-SIIW$_R$** | 100% | 0.26 min | 1.21 min | 1.47 min | 28 | 4 |
| | | | | | PDDLStream *Adaptive* | 100% | 0.03 min | 1.24 min | 1.27 min | N/A | 1 |
| | | | | | PDDLStream *Binding* | 100% | 0.21 min | 1.20 min | 1.41 min | N/A | 1 |
| | | | | | PDDLStream *Incremental* | 100% | 12.01 min | 1.22 min | 13.23 min | N/A | 1 |
| | | | | | PDDLStream *Focused* | 0% | N/A | N/A | N/A | N/A | N/A |
| | | 25 | 5 | Med. | **Lazy-SIIW$_R$** | 100% | 3.36 min | 3.95 min | 7.31 min | 235 | 10 |
| | | | | | SIW$_R$ | 100% | 21.60 min | 4.02 min | 25.62 min | 198 | 10 |
| | | | | | BFWS | 100% | 13.47 min* | 7.61 min | 21.08 min | 3.5k | 1 |
| | | | | | PDDLStream (Any) | 0% | N/A | N/A | N/A | N/A | N/A |
| | 4 | 7 | 7 | Low | **Lazy-SIIW$_R$** | 100% | 2.35 min | 4.45 min | 6.80 min | 185 | 14 |
| | | | | | Planet | 100% | 3.22 min | N/A | N/A | N/A | 1 |
| | | | | | PDDLStream *Adaptive* | 100% | 7.47 min | 5,72 min | 13.19 min | N/A | 1 |
| | | | | | PDDLStream (Others) | 0% | N/A | N/A | N/A | N/A | N/A |
| | | 28 | 14 | Med. | **Lazy-SIIW$_R$** | 100% | 6.52 min | 9.24 min | 15.76 min | 630 | 34 |
| | | | | | Planet | 0% | N/A | N/A | N/A | N/A | N/A |
| | | | | | PDDLStream (Any) | 0% | N/A | N/A | N/A | N/A | N/A |
| Non-Mono. | 2 | 10 | 10 | Low | **Lazy-SIIW$_R$** | 100% | 3.53 min | 8.10 min | 11.63 min | 565 | 30 |
| | | | | | SIW$_R$ | 100% | 17.69 min | 7.96 min | 25.65 min | 534 | 30 |
| Words | 1 | 11 | 4-5 | Low | **Lazy-SIIW$_R$** | 100% | 2.26 min | 3.49 min | 5.75 min | 243 | 13 |
| | | | | | SIW$_R$ | 100% | 8.07 min | 3.58 min | 11.65 min | 216 | 13 |

Table 2: Profiling of the action-validation pipeline.

| Step / Function | Success | Timeout | Rel. succ. | Cum. succ. |
|---|---|---|---|---|
| 1. *InArmWorkspace* | 35 µs | N/A | 5-30% | 5-30% |
| 2. *InverseKinematics* | 9 ms | 0.15 s | 20-40% | 1-10% |
| 3. *MotionPlan* | 0.38 s | 5 s | 80-100% | 0.5-10% |

space where the obstructing objects can be set aside without blocking misplaced objects. In this scenario, most of the inverse-kinematic (IK) computations, which are an iterative process, reach the IK user-provided timeout without finding a solution. Thus, the action is marked as unfeasible (even when it is valid, simply because it has not had time to find a motion that implements it). This implies a slower node expansion (it is more probable to spend the whole IK time-budget in an expansion since there is not an actual solution) and a higher difficulty to find valid actions (i.e. longer total planning time). Note that the selection of the IK time-budget is not trivial: If its extremely low, IW(1) could discard all the possible grounded actions that can lead to the goal (risking not being able to find a solution plan even when one exists) and, on the contrary, being too high implies non-affordable computational times. In this approach, 5 seconds has experimentally been found to keep a trade-off between finding solutions even in cluttered environments maintaining a reasonable computational-time. In addition, as there are so few gaps, the proposed approach samples the placements more densely to ensure that there exists an accessible placement. On the contrary, PDDLStream planners and Planet do not scale well with the number of objects. The BFWS search also scales well but at the cost of a rigid precompilation that forces the objects to be in a fixed set of grid configurations only during the search. This precompilation is expensive and does not scale up that well, but when it does so, the planner does not need to invoke the motion planner at plan time.

## Conclusions

We have presented an approach for TAMP that makes use of the notions of width and sketches developed in classical planning for decomposing problems into subproblems. For this, a general sketch of width 1 has been crafted that ensures that the families of pick-and-place problems considered are decomposed into subproblems that can be solved greedily in linear time, under suitable assumptions. The same sketch has been used for the three types of tasks considered; the only change being in the definition and computation of one of the features (number of misplaced objects). The value of sketches is that they allow to specify the subgoal structure of the problems at a high-level, without having to specify unnecessary details that are handled by the polynomial, and in our case, linear search. The language of sketches is very flexible and offers two clear benefits in the context of TAMP. First, when a task plan is found to be unfeasible due to the geometric constraints, the combinatorial search resumes in the specific subproblem where unfeasible subplan was found. Second, the sampling of object poses is not done once, globally, at the start of the search, but locally, at the start of each subproblem. The bounded width of the sketch provides a bound on the complexity of the subproblems and an (approximate) guarantee that only shallow backtracks will be needed (in subproblems but not across subproblems). The proposed approach has been integrated within the ROS environment, and will be made available.

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
