# OpenReview forum: "Combined Task and Motion Planning Via Sketch Decompositions"
_icaps-conference.org/ICAPS/2024/Conference — ICAPS 2024_

### Official Review · Reviewer_8nBH · 2024-01-02

**Significance And Importance:** 2
**Soundness:** 3
**Novelty:** 3
**Clarity:** 3
**Overall Evaluation:** 2
**Confidence:** 3

**Weaknesses:**

1: Minor weaknesses that are easily fixable.

**Contributions Of The Paper:**

This paper proposes a way to solve problems that involve both motion and task planning, e.g. a robot acting in continuous 3d space to manipulate blocks on tables into various configurations. The domain is transformed into a classical planning problem by sampling the continuous parameters, and a set of features are computed from the variables that define the domain. A human human crafted policy sketch over the features is used to decompose the problem into a series of subgoals, which are solved one after the other using search algorithms based on width. Dedicated motion algorithms are used to verify the individual steps of the solution, and the discretization is further refined if search at any one of the steps fails.

I believe the discretization + classical planning approach for combined motion and task planning has previously been proposed, as has the features + policy sketches + iterated width approach for solving classical planning problems. The combination of the two to solve these types of problems by using dedicated algorithms to check the motion component of the individual steps is new. The proposed algorithm seems to be a good match to the task, as the task planning component, once decoupled from the motion planning component, is fairly simple and intuitive.

**Ethical Considerations:**

(1) Not Applicable: The paper does not have any ethical considerations to address

**Nomination For Best Paper:**

No

**Questions For Authors:**

In the "Action Schemas" section, you refer to the set of grasps g and and stable object poses \sigma. Are these explicit problem variables or are they implicit based on the the 3 dimensional continuous variables that define the configuration of the robot / objects and only used as variables here to make the concepts clearer to the reader?

Is it necessary to create sampled groundings for every action in the domain during each subgoal search? Could the space somehow be pruned to create finely sampled groundings of only those actions that are potentially related to the subgoal being considered?

line 221: "The parameter values are randomly sampled" what is the benefit of randomness here, as opposed to interpolating evenly spaced values between those that need to be accommodated? (ie the values in the current state s + the goals)

If I understand correctly, a problem similar to towers of hanoi could not be solved by this method since there are might not be available slots to temporarily discard a block / disc. If so maybe worth pointing out as an easy to recognize example of a domain that does not satisfy the assumptions made.

minor:

line 187: what does SE stand for? would be helpful to explain this notation.

line 189: "homogeneous transformations" does this just mean that the possible transformations are the same for the robot and each of the objects? this could be clearer.


typos etc.

line 72: All the tasks considered as benchmark*s*

"a goal that fits to many robotic manipulation tasks" remove 'to'

line 126: s/leaving/placing/ to avoid ambiguity

line 144: "forward from the goal" from the initial state?

"a standard on this field." on --> in

"inspired in" --> inspired by

"The action is not be feasible" remove 'be'

"capture the robot and objects poses" add possessive apostrophe, "objects' poses"

algorithm carries *out* an IW search

"During the search for such a minimum value, it is considered the best robot location" rephrase, it is considered is ambiguous (something is considered the best location, vs the best robot location is considered)

line 366, s/F/I/ ? if not explain F

which would not block *an*other misplaced object

line 452: will not lead *to* a state

Since the sketches allows --> allow

the novelty management is adapted such --> s/such/as follows/

**Reproducibility:**

3: Authors describe the implementation and domains in sufficient detail.

**Strengths Of The Paper:**

The proposed method is an interesting combination of previous approaches and seems to be quite effective. It provides a way to leverage a robust form of classical planning for robotics problems.

The paper is overall a good read and well presented (though some sections would benefit from rewriting and thinking about how to simplify the explanations, more below)

**Weaknesses Of The Paper:**

Overall the paper is well written, but there are a few parts where there are missing words, sentences are confusing, or the explanations could be improved.

The definitions of u and v are confusing and would benefit from breaking down \alpha and \beta and formalizing their explanations a bit more. The range for the sums and minimums need to be clearer, as well as when a value is defined for a particular object i and when it is over all objects.

The definitions of the sketch conditions are confusing and would benefit from being uniform, e.g. "if x, do y". currently the last rule has "enforcing to put it down" instead of just "put it down" for example. It's not clear what "if it had a goal associated with it" refers to. The notes relating to syntax (e.g. this notation means change but not increase) could be moved outside of the bullet list so these are more concise.

This is just personal preference, but I think the rules would be clearer if they specified the features that have to remain the same and omitted features which are allowed to change freely.

---

> ### Author Rebuttal · Authors · 2024-01-28
>
> We thank the reviewer for valuable comments and suggestions. We will improve the explanations for the camera-ready paper.
> - In u and v definitions, the range of the minimum and sum are over all the misplaced objects.
> - We used the sketch notation of the work that first introduced the sketches (Bonet and Geffner, 2021).
>
> Q1
> The grasps and SOPs are explicit problem variables. SOPs depend on the problem object, e.g. a cube does not stand up in the same ways as a cylinder. The grasps depend on the problem object and the robot, e.g. a cube can only be picked up laterally from one of its 4 faces, whereas a cylinder has infinite faces. Moreover, a suction cup gripper offers different grasps than a parallel finger gripper.
>
> Q2
> During the sub-search, we only ground those actions that have as initial state the one of the node being expanded. The child nodes that do not pass the novelty filter are discarded (along with all their offspring). Also, the proposed sketch generates sub-problems with very short solutions (i.e. mostly only 1-step long). Thus, the number of grounded actions is very small, especially compared to other approaches (see Table 1).
>
> Q3
> Random parameter sampling is computationally advantageous as it efficiently explores diverse solution spaces, mitigating the curse of dimensionality by adapting to high-dimensional environments, avoiding local minima, and handling complex geometries more effectively than deterministic interpolation. Besides, using random (and incremental, since it is densified with each failed sub-search) sampling means that with enough time the algorithm ends up sampling parameter values that do allow the discrete problem to have a solution, i.e. the algorithm is probabilistically complete.
>
> Q4
> The proposed sketch will fail for the Hanoi domain, since it relies on the assumption of having enough space to place the objects without causing new interferences. However, the approach is still valid but the sketch would need to be modified to account for this requirement.
>
> Q5
> SE is the special Euclidean group, which comprises arbitrary combinations of translations and rotations, but not reflections. We will clarify in the camera-ready paper.
>
> Q6
> The sets of homogeneous transformations (i.e. combination of translations and rotations) considered for the robot and the objects are disjoint, since the robot must always stand on the floor and the objects must stand on the table or within the robot gripper. We clarify in the camera-ready paper.

---

### Official Review · Reviewer_SnN9 · 2024-01-19

**Significance And Importance:** 2
**Soundness:** 4
**Novelty:** 3
**Clarity:** 4
**Overall Evaluation:** 2
**Confidence:** 5

**Weaknesses:**

1: Minor weaknesses that are easily fixable.

**Contributions Of The Paper:**

The authors propose a novel interleaved approach that utilizes sketch decomposition in combined task and motion planning. Moreover, their approach can solve problem instances that may be deemed unfeasible due to geometric constraints. The experiment with the current state of the art demonstrated that the proposed approach yielded better scalability and planning time results.

**Ethical Considerations:**

(1) Not Applicable: The paper does not have any ethical considerations to address

**Nomination For Best Paper:**

No

**Questions For Authors:**

1. The paper mentions that the ROS environment will be made available. Will this be possible before to include right now?

2. The paper mentions that PDDLStream/Planet does not scale well. Could you elaborate more in the paper?

3. (Line 203) States that the robot can perform the action Pick if the gripper is not occupied. What if the object is missing due to the robot dropping it? How would it handle these scenarios and other potential anomalies, such as one object blocking another?

**Reproducibility:**

4: Authors promise to release code and domains (whichever apply).

**Strengths Of The Paper:**

The authors cover the related work extensively, and the paper is well-written and presented. The authors justified the advantages of their approach and the need for it in TAMP. They tackle a classic yet challenging TAMP problem. The results demonstrated the need for their approach as it improves the current state of the art.

**Weaknesses Of The Paper:**

Overall, the paper is excellent and does not contain a lot of weaknesses. However, adding other problems to demonstrate the scalability of this approach would be beneficial as it solves the problems in the experiment fairly quickly compared to the benchmarks.

The paper does not mention future work, which would be nice to see.

---

> ### Author Rebuttal · Authors · 2024-01-28
>
> Weaknesses:
>
> We thank the reviewer for valuable comments and suggestions.
> - Regarding scalability, we focused on showing scalability for the sorting objects task, which is the one for which comparison with alternative methods was not too difficult. As shown theoretically, we expect linear scalability for all the solved problem families (i.e.Sorting Objects, Non-monotonic and Words), since the sketch has width 1 an it decomposes the problem into subproblems that can be solved greedily in linear time.
> - Regarding the future, we did our best given the space limits but we will try to include it in the camera-ready version of the paper. Interesting topics that could be treated are: 1) The definition of sketches to tackle new problem families and the inclusion of learning AI-based techniques into the proposed approach to obtain self-learned rules, features and sampling strategies. 2) The extension of the proposed approach to consider optimality and being able to tackle problems with state uncertainty and partial observability, where dynamics- or physics-based planning is needed. 3) Post-processing and smoothing of the obtained plans by using optimal motion-planning algorithms (e.g. Informed RRT∗) and merging and shortening the trajectories of consecutive actions.
>
> Question 1:
> We will release all the source code once the paper is accepted. Our institutions do not allow us to release the source code before. We will also try to add our contribution to the PPDL stream repository.
>
> Question 2:
> As shown in Table 1, the performance of these approaches significantly downgrades as the number of problem objects increases, sometimes being unable to solve the problem. It is mainly explained by the fact that these approaches do not serialize the problem into smaller subproblems which allow resuming failed searches in a specific subproblem and to tailor the automatic sampling of the continuous world to the specific subproblem.
>
> Question 3:
> Everytime the Pick action is checked it is for an object that is not missing. If the object has been dropped it will consider its actual position and plan accordingly. The scenarios such as one object blocking another are handled by the proposed method and appear in the three benchmark tasks. When this situation arises, a subgoal is generated that implies removing the blocking object(s).

---

### Official Review · Reviewer_4mkL · 2024-01-19

**Significance And Importance:** 3
**Soundness:** 3
**Novelty:** 3
**Clarity:** 3
**Overall Evaluation:** 1
**Confidence:** 3

**Weaknesses:**

2: No major or minor weaknesses.

**Contributions Of The Paper:**

The paper presents a TAMP method based on sketches, which is a method to decompose problems into smaller problems. The method employs hand-made sketches to search for subproblems, which are then solved using the IW(1) algorithm. The experimental evaluation compares with different available approaches and shows efficiency improvements.

**Ethical Considerations:**

(4) Good: The paper adequately addresses most, but not all, of the applicable ethical considerations

**Nomination For Best Paper:**

No

**Questions For Authors:**

- The presented sketches work well for pick-and-place scenarios and are generalisable in those contexts, but how extensible is the method to other TAMP tasks involving other kinds of motions?

**Reproducibility:**

4: Authors promise to release code and domains (whichever apply).

**Strengths Of The Paper:**

- The proposed approach seems efficient and the authors propose some other efficiency extensions.
- The claims of the paper are proved.
- The authors present a real robotics scenario.
- The code will be made available.

**Weaknesses Of The Paper:**

- While the paper is quite clear, it could benefit from some description of the methods used instead of assuming knowledge of them, to make it more self-contained.
- In line 289, the CMTP acronym is used but not defined.
- It is unclear whether the success ratio in Table 1 includes execution. If not, the paper would benefit also from some execution metrics.
- The authors mention they use the motion planner MoveIt, however, MoveIt is not a motion planner but a framework to interface with different motion planners.
- The video was not available (even after retyping the URL, it said the file did not exist).

---

> ### Author Rebuttal · Authors · 2024-01-28
>
> Weaknesses:
>
> We thank the reviewer for valuable comments and suggestions.
> - Regarding adding a more description of the methods used, we could add additional descriptions of the methods we use for comparing our approach, in the appendix, due to the space limitations.
> - CMTP is a typo. It should be CTMP (Combined Task and Motion Planning), another acronym to refer to TAMP. We will correct this in the revised version.
> - Success ratio in Table 1 is always 1, since the plan includes not only the high-level plan but also the low-level motions
> - We agree that MoveIt is a framework, not a motion planner. We will change the text accordingly
> - We have corrected the link and should work now: https://we.tl/t-CFFWrKLMzX
>
>
> Question 1: The method is general but the presented sketch is designed for pick-and-place scenarios. For TAMP tasks involving other kinds of motions such as drag, push, etc., the sketch can be reused, but new features would need to be implemented accordingly.

---

### Meta-Review · Area_Chair_UZzd · 2024-02-06

**Recommendation:** Accept (Oral)
**Confidence:** 4

**Metareview:**

The paper presents a method for task and motion planning (TAMP) based on sketches. The sketches decompose the problem into a sequence of subgoals. The efficiency is demonstrated on pick-and-place scenarios with a hand-written sketch of width 1, thus the subproblems can be solved greedily in linear time. One reviewer has concerns regarding the extensibility of the sketches to other scenarios but all reviewers are overall happy with the quality, clarity, originality and significance of the work and recommend to accept the paper.

Pros:
- approach is experimentally efficient
- interesting combination of previous approaches
- overall well-written

Cons:
- one reviewer has concerns regarding the extensibility of the sketches to other scenarios
- presentation should be improved in some specific minor aspects

**Ethical Considerations:**

(1) Not Applicable: The paper does not have any ethical considerations to address